environmental science/environmental engineering/power and energy systems

gaseous streamer corona plasma, aniline, degradation, spectrum analysis, water treatment

**Author for correspondence:**
Anna Zhu
e-mail: zhuanna1234@126.com

# Degradation of aniline in water with gaseous streamer corona plasma

Yang Li[1], Zhanguo Li[1], Zhen Liu[2], Shitong Han[1], Sanping Zhao[1], Keping Yan[2] and Anna Zhu[1]

[1]State Key Laboratory of NBC Protection for Civilian, Beijing 102205, People's Republic of China
[2]Industrial Ecology and Environment Research Institute, Zhejiang University, Hangzhou 310000, People's Republic of China

YL, 0000-0002-0296-8563

This paper demonstrated the effects and influencing factors in degrading aniline by gaseous streamer corona plasma along water surface under different discharging gas atmospheres. For aniline with an initial concentration of $100 \, \text{mg} \, \text{l}^{-1}$, the degradation was fastest when the reactor was not ventilated, and the degradation rate is 98.5% under 7.5 min treatment. While the degradation was slowest when Ar was ventilated, the degradation rate is 98.6% after treatment for 60 min. Some active particles were detected using a multi-channel fibre-optic spectrometer during the discharge, such as Ar, OH, $N_2$, $N_2^+$ and N. In particular, NO was detected during air discharge. The NO and $N_2^+$ could produce $NO_3^-$; then generated nitric acid would affect the pH value of the solution. The intermediate product by $N_2$ discharge is nitrophenol, and nitrophenol would be converted to *p*-benzoquinone. The $O_2$ discharge could produce an intermediate product of aminophenol. The intermediate products in Ar discharge were in small amounts and the final mineralization effect was the best.

## 1. Introduction

Aniline is widely used as one of the most important intermediates in many industries, such as textile and petroleum [1]. Aniline is a kind of toxic organic pollutant that is harmful to the environment and humans; therefore, it is in the priority pollutants list of the US Environmental Protection Agency (EPA) [2]. In recent years, emergency accidents caused by aniline leakage have occurred frequently. These accidents, especially when aniline leaks into the water environment, always cause serious environmental problems and threaten human health [3]. Several methods have been employed for the treatment of aniline, such as bioremediation [4–6], oxidation [7,8] and adsorption [9–11]. For

R. Soc. Open Sci. **8**: 203314

**Figure 1.** Reactor structure and discharge image.

the bioremediation technology, it has the advantage of economic and environmentally friendly, but requires a long processing time. The oxidation and adsorption methods may be simple and effective, but usually have the problem of causing secondary pollution or are subjected to secondary treatment. In recent years, advanced oxidation processes (AOPs) have received much attention for the degradation of aniline due to the generation of reactive species [12–14]. As a kind of AOP, the low-temperature plasma generated by high-voltage pulse discharge is an effective way to degrade contaminants, and the above disadvantages are avoided [15–17]. Various discharge methods have been used for plasma generation, such as discharge in liquid phase, gaseous discharge and discharge at the water surface [18]. Gaseous discharge is considered to create large volumes of plasma under ordinary conditions [19], and it is beneficial to the degradation of pollutants. Different active species will be produced when discharging under different gases atmosphere, resulting in different degradation effect of pollutants [20–23]. Here, we investigated the effects and influencing factors in degrading aniline in water by pulsed streamer corona plasma along water surface under different gas atmospheres, such as air, oxygen, argon and nitrogen. The active specie and the intermediate products generated in the discharging processes were analysed with multi-channel fibre-optic spectrometer and high-performance liquid chromatography (HPLC).

# 2. Material and methods

## 2.1. Experimental materials

The reagents used in this experiment were aniline (analytical grade, Tianjin Zhiyuan Chemical Reagent Co., Ltd), *p*-aminophenol, *o*-aminophenol, *p*-benzoquinone, *p*-nitrophenol and *o*-nitrophenol (analytically pure, Tianjin Guangfu Fine Chemical Research Institute), *m*-nitrophenol (analytical grade, Aladdin Reagent Shanghai Co., Ltd) and nitric acid (analytical grade, Hangzhou Gaojing Fine Chemical Co., Ltd).

## 2.2. Experimental method

A needle-plate reactor used in the experiments is shown in figure 1*a*. The reactor consists of the following components: 1, a stainless steel high-voltage pole, connected with a high-voltage pulse; 2, a gas pipe, used for injecting different gases; 3, a Plexiglas cover; 4, a gas mixing chamber; 5, a metal joint, where a metal needle can be inserted; 6, a Plexiglas reactor tube; 7, a metal disc ground electrode; and 8, a stainless steel needle, used as the high-voltage discharge electrode.

In the experiments, 200 ml aniline solution with an initial concentration of 100 mg l$^{-1}$ was added into the reactor. The rod-shaped high-voltage pole was connected to the high-voltage pulse power source, and the aniline solution was degraded by the gaseous streamer corona plasma. The discharge frequency was set to 1000 Hz, and the voltage was 20 kV. A typical pulse voltage and current waveform is shown in figure 1*b*, and the gaseous streamer corona plasma image is shown in figure 1*c*. The number of discharging needles was 12, and the distance between discharging needles tips and water surface was 5 mm. The total treatment time was 60 min. The aniline concentration and its degradation products were analysed by an Agilent 1260 HPLC equipped with an Eclipse Plus C18 column (4.6 × 100 mm, 5 µm).

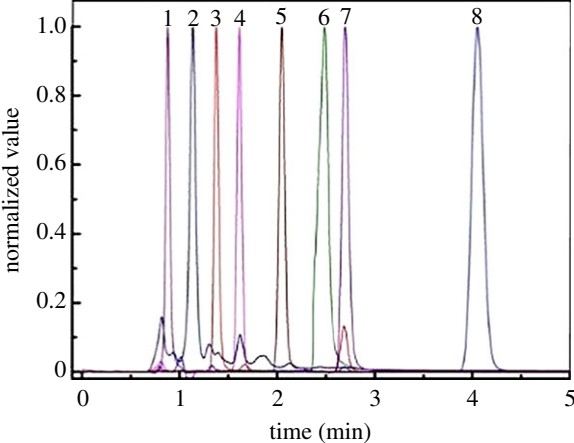

**Figure 2.** HPLC peaks of the target analytes.

The HPLC method was set as follows: the column temperature was 25°C, the mobile phase was pure water (60%) and acetonitrile (40%), the flow rate was 1 ml min$^{-1}$ and the detection wavelength was 242 nm. The sampling times during degradation process were 0, 2.5, 5, 7.5, 10, 12.5, 15, 17.5, 20, 30, 40, 50 and 60 min.

This study investigated the degradation of aniline solution under different gases atmospheres. The gas was introduced into the gas pipe and mixed completely in the gas mixing chamber, and then was ejected from the high-voltage electrode needle. The gas flow was set at 0.5 m$^3$ h$^{-1}$. There was a small opening in the wall of the reactor to allow the gas to be exhausted. The degradation of aniline was investigated under different gas atmosphere as follows: 1, no gas venting; 2, venting with air; 3, venting with O$_2$; 4, venting with Ar; and 5, venting with N$_2$. It should be noted that no gas venting means no gas flow, but still have ambient air.

In the experiments, an Avantes Avanspel-2084 multi-channel fibre-optic spectrometer was used to detect the emission spectra of the active substances generated during discharging processes under different gas atmospheres.

The energy density, $E$, is defined as shown in the below formula

$$E = \frac{Pt}{V},\tag{2.1}$$

where $P$ is the plasma discharge power, which is calculated as previously reported [19], $t$ is 60 min and represents the discharge time, and $V$ is 0.2 l, representing the volume of the treatment liquid.

## 2.3. Determination of aniline and its intermediate products

Aniline and its possible intermediate products, such as $p$-aminophenol, $o$-aminophenol, $p$-benzoquinone, $p$-nitrophenol, $m$-nitrophenol, $o$-nitrophenol and nitric acid, were detected by HPLC. The chromatographic peaks of each pure standard solution are shown in figure 2, where 1, nitric acid; 2, $p$-aminophenol; 3, $o$-aminophenol; 4, $p$-benzoquinone; 5, aniline; 6, $p$-nitrophenol; 7, $m$-nitrophenol; and 8, $o$-nitrophenol. The retention times and the equations used to calculate the concentration of each substance are shown in table 1.

# 3. Results and discussion

## 3.1. Degradation of aniline in water

Degradation of aniline in water by gaseous streamer corona plasma under different experimental conditions was investigated. Samples were taken at different times and analysed by HPLC. The results of aniline degradation are shown in figure 3$a$, in which the results of experiments conducted with no gas ventilation are used as the control. It can be seen from figure 3$a$ that aniline in water is effectively degraded under different experimental conditions.

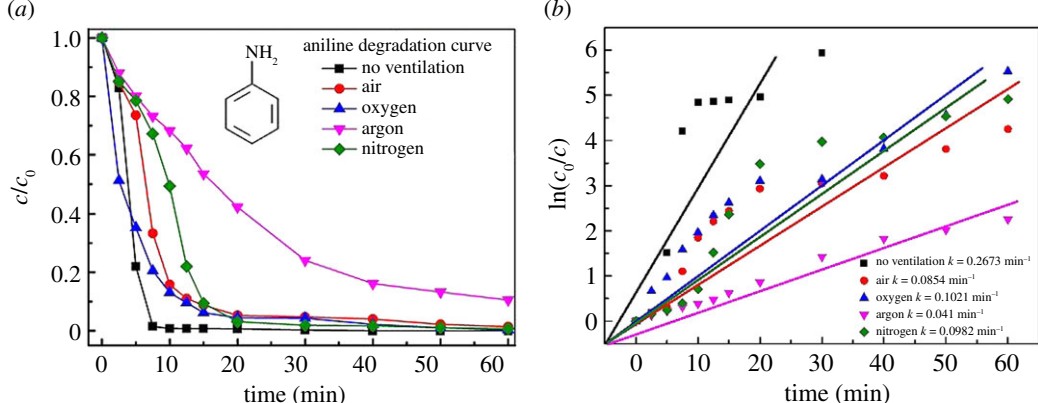

**Figure 3.** Aniline degradation curves of different gases.

**Table 1.** HPLC retention times of the target analytes. $A$, the peak area; $c$, the concentration (mg $l^{-1}$).

| serial number | substance | retention time (min) ($\pm 0.01$) | standard curve |
|---|---|---|---|
| 1 | nitric acid | 0.88 | |
| 2 | *p*-aminophenol | 1.14 | $A = 34.601c - 13.129$, $R^2 = 0.9982$ |
| 3 | *o*-aminophenol | 1.38 | $A = 16.481c - 44.719$, $R^2 = 0.9968$ |
| 4 | *p*-benzoquinone | 1.61 | $A = 114.03c$, $R^2 = 0.9978$ |
| 5 | aniline | 2.05 | $A = 32.347c + 9.1729$, $R^2 = 0.9998$ |
| 6 | *p*-nitrophenol | 2.49 | $A = 8.7094c - 48.711$, $R^2 = 0.9963$ |
| 7 | *m*-nitrophenol | 2.70 | $A = 18.102c$, $R^2 = 0.9998$ |
| 8 | *o*-nitrophenol | 4.05 | $A = 4.8872c - 10.542$, $R^2 = 0.9988$ |

Aniline can be almost completely degraded in 7.5 min under no gas ventilated condition (the degradation rate was 98.5%), and the degradation efficiency of aniline becomes lower when the system is ventilated with gases. On the one hand, the gas flow influences the time that active species contact water surface. On the other hand, the gas flow influences the $O_3$ generation [24], the main function of $O_3$ is to cleave benzene rings [25]. So the degradation of aniline is better when the system is not ventilated, and the gas flow will adversely affect discharge degradation. When $O_2$ was used for ventilation, the degradation of aniline was better than air ventilation condition. This is because the $O_2$ content is 100% when pure $O_2$ is passed through the system, while $O_2$ accounts for 21% in the air. According to Lukes *et al.* [24], the greater the proportion of $O_2$ in the gas, the higher the concentration of $O_3$ in the discharge, so the degradation effect when ventilating with pure $O_2$ is better than when ventilating with air. When $N_2$ and Ar are introduced, there is no $O_2$ and the possibility of generating $O_3$ is excluded. Under these conditions, the degradation of aniline mainly depends on the active substances generated in the discharging processes. The active substances dissolve in the water to generate hydroxyl radicals and hydrogen peroxide could also degrade the aniline simultaneously. As the active substances was relatively simple (detailed in §3.3), the degradation rate of aniline is lower at the beginning in the case of Ar and $N_2$ discharging. According to the experimental results, it was preliminarily concluded that the main factor affecting the degradation of aniline was $O_3$ dissolved in water. The degradation process basically followed the first-order kinetic reaction equation, as shown in figure 3*b*. The degradation rate constant of no ventilation was 0.267 $min^{-1}$ and Ar discharge was 0.041 $min^{-1}$.

## 3.2. The intermediate products during the degradation of aniline

A series of intermediate products in the processes of degrading aniline with gaseous streamer corona plasma were detected by HPLC, including *p*-aminophenol, *o*-aminophenol, *p*-benzoquinone, *p*-nitrophenol, *m*-nitrophenol and *o*-nitrophenol. These intermediates vary in their behaviour when

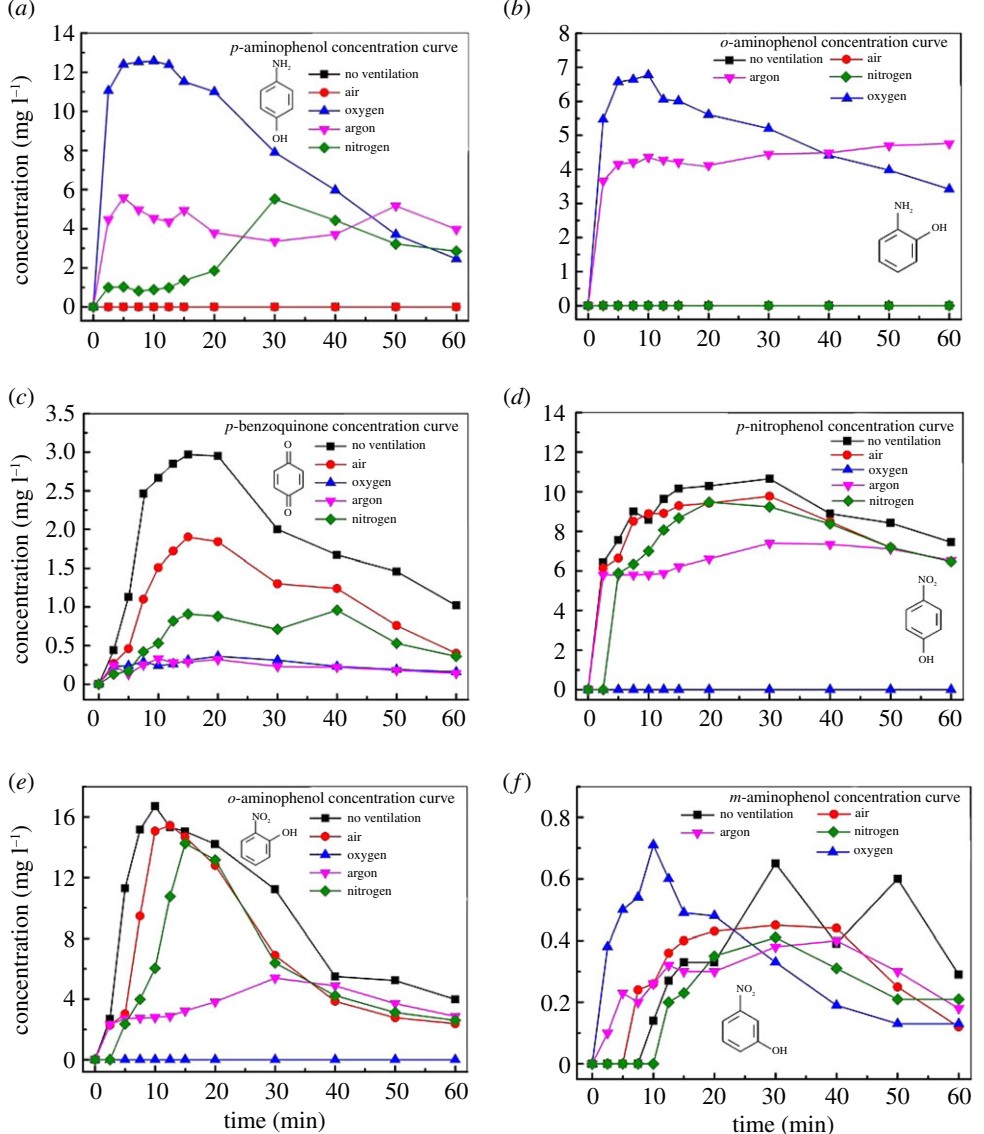

**Figure 4.** The intermediate products during the degradation processes of aniline under different discharge gas atmosphere.

different gases are ventilated, but the general trend shows that some of the aniline is converted to these products and then oxidatively degraded along with the aniline. As shown in figure 4a, when $O_2$ is introduced, the $p$-aminophenol concentration is the highest, and when Ar and $N_2$ are introduced, the $p$-aminophenol concentration is low. When the air is introduced or no ventilation, $p$-aminophenol was not detected. Figure 4b shows that $o$-aminophenol is formed when $O_2$ and Ar are introduced. When $N_2$ or air were introduced or no ventilation, $o$-aminophenol was not detected. In figure 4c, the amount of $p$-benzoquinone is the highest when the system is not ventilated, and it is extremely low when $O_2$ and Ar are introduced. As shown in figure 4d,e, no $p$-nitrophenol and $o$-nitrophenol are detected in the solution when $O_2$ is introduced. When Ar is introduced, the amount of $p$-nitrophenol and $o$-nitrophenol produced is less than that produced by ventilation with air or $N_2$ or when there is no ventilation. A trace amount of $m$-nitrophenol is also detected during the discharge degradation process, as shown in figure 4f.

It can be seen from the above analysis that in the degradation processes involving oxidation, aminophenol tends to form, as shown in figure 4a,b. When the reactor is ventilated with oxygen, the concentration of $p$-aminophenol and $o$-aminophenol is relatively high. As shown in figure 4d,e, $p$-nitrophenol and $o$-nitrophenol are not formed when the reactor is ventilated with oxygen. As shown in figure 4f, the concentration of $m$-nitrophenol is also very low when oxygen is supplied. It can also be seen that when degrading aniline under $N_2$ atmosphere, more nitrophenol is formed. As shown in

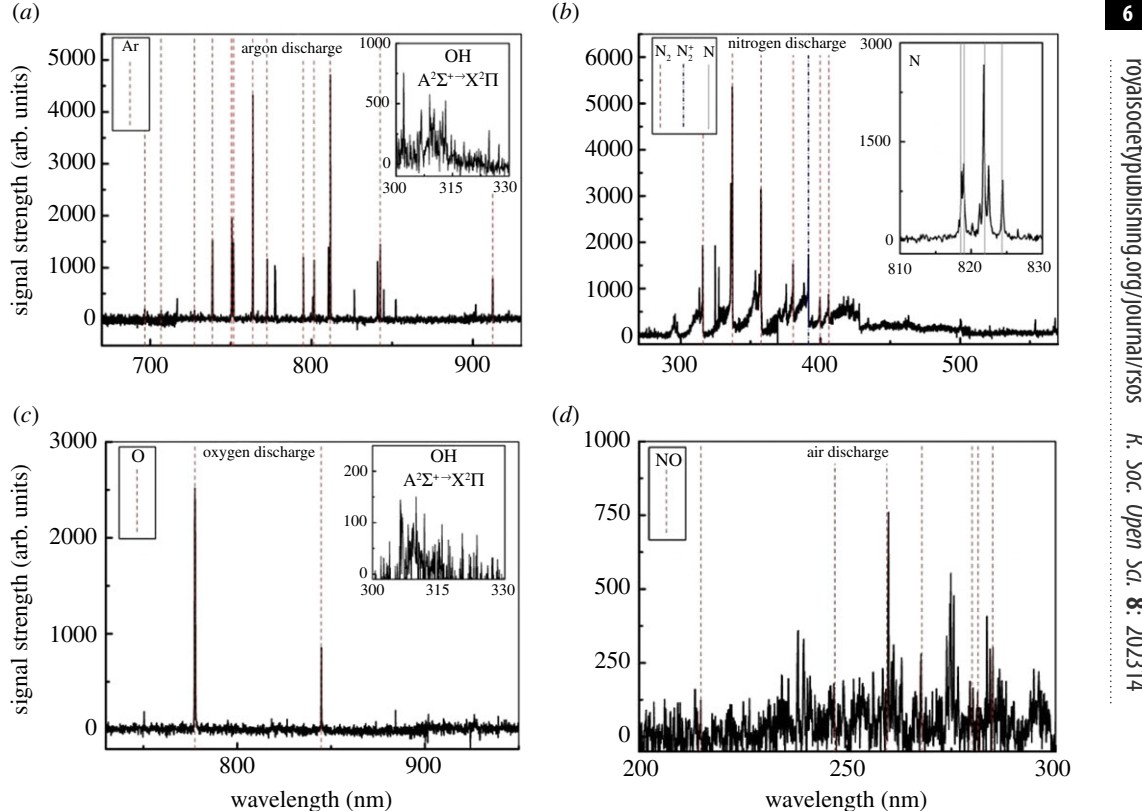

**Figure 5.** Emission spectra under different discharging atmospheres.

figure 4*d*–*f*, *o*-nitrophenol, *m*-nitrophenol and *p*-nitrophenol are formed when the reactor is ventilated with $N_2$. As $N_2$ accounts for approximately 80% of the content of air, nitrophenol is also formed under no ventilation or when the reactor is ventilated with air. It can be seen from figure 4 that when Ar is introduced, the aminophenol and nitrophenol concentrations are between those generated in the processes under $O_2$ and $N_2$ atmosphere.

## 3.3. Emission spectrum analysis

### 3.3.1. Ar discharge emission spectroscopy

In the Ar discharging processes, the high-energy electrons collide with the Ar atoms to generate excited Ar atom, and the excited Ar atoms can react with water to form OH and H radicals, as shown in the below formulae [20,26]

$$e^* + Ar \rightarrow Ar^* + e \tag{3.1}$$

and

$$Ar^* + H_2O \rightarrow Ar + OH + H. \tag{3.2}$$

The emission spectrum of the Ar atom is in the range of 690–920 nm, as shown in figure 5*a*. The position of these lines can be determined as 696.54, 707.62, 727.29, 738.40, 750.39, 751.47, 763.51, 772.42, 794.82, 801.47, 811.53, 842.46 and 912.30 nm, which are sequentially marked with red dashed lines [27,28]. The upper state energy of these excited Ar atoms is approximately 13 eV. The OH ($A^2\Sigma^+ \rightarrow X^2\Pi$) radicals with wavelengths ranging from 306.1 to 330 nm were also detected in the emission spectrum [29].

### 3.3.2. $N_2$ discharge emission spectroscopy

In the case of $N_2$ discharge, the high-energy electron collision with ground state $N_2$ can generate excited state $N_2$ ($C^3\Pi_u$). The attack of high-energy electrons on $N_2$ can also generate excited state N atoms. The

process is as shown in the below formulae [30]

$$e^* + N_2 \rightarrow N_2(C^3\Pi_u) + e \tag{3.3}$$

and

$$e^* + N_2 \rightarrow N^* + N^* + e. \tag{3.4}$$

The emission spectrum of the $N_2$ discharge is shown in figure 5b. The second positive transition line of $N_2$ ($C^3\Pi_u \rightarrow B^3\Pi_g$) was detected during discharge. The positions are 316.0, 337.1, 357.5, 380.5, 399.8 and 405.9 nm, which are sequentially marked with red dashed lines [31]. The line of $N_2^+$ ($B^2\Sigma_u^+ \rightarrow X^2\Pi_g^+$) is also detected at 391.44 nm and is indicated by a blue dashed line [32]. The spectral line positions of the N atoms are detected at 818.5, 818.8, 822.3 and 824.3 nm, which are sequentially indicated by solid grey lines.

### 3.3.3. $O_2$ discharge emission

In the process of $O_2$ discharge, due to the low dissociation energy of oxygen molecules, it is easy to dissociate and generate excited O atoms after collision with high-energy electrons. The formula is as follows:

$$e^* + O_2 \rightarrow O^* + O^* + e. \tag{3.5}$$

The emission spectrum of the oxygen discharge is shown in figure 5c. The position of the O atom spectral lines can be determined as follows: 777.4 and 844.6 nm, which are indicated by red dotted lines [33]. As shown in figure 5c, the OH ($A^2\Sigma^+ \rightarrow X^2\Pi$) radical spectrum with a wavelength in the range of 306.1–330 nm was detected during the discharge process, which was caused by the impact of high-energy electrons on gaseous water vapour, as shown in the below formula [34]

$$e^* + H_2O \rightarrow OH + H + e. \tag{3.6}$$

### 3.3.4. Air discharge emission

The composition of air is more complicated, the main components of which are $N_2$ and $O_2$. When discharging in air, in the spectrum of emission, in addition to a series of products the same as that in $N_2$ and $O_2$ discharge are detected, the spectral line of NO ($A^2\Sigma^+ \rightarrow X^2\Pi$) is also detected. Figure 5d is an emission spectrum of NO during air discharging. The NO wavelength ranges from 200 to 300 nm [35,36], the position of the spectral lines are as follows: 214.8, 247.0, 259.5, 267.9 , 279.9, 280.9 and 284.9 nm, which are marked with red dashed lines. This indicates that in the air discharge, NO is generated due to the interaction of $N_2$ and dissociated O from oxygen gas, as shown in the below formula [37]

$$N_2(A^3\Sigma_u^+) + O^* \rightarrow NO + N^*. \tag{3.7}$$

## 3.4. Changes in solution parameters

At the beginning and the end of treatment, the aniline solution is sampled and tested for its pH, conductivity, temperature and TOC. The results are shown in figure 6. After treatment for 60 min, the pH of the solution significantly reduced under all the discharging atmospheres, which means the solution becomes acidic. It is estimated that the aniline ring is broken to form carboxylic acids [38], or nitric acid is formed in the solution during the degradation of aniline. Nitric acid can be formed during air discharge and nitrogen discharge because NO produced in the discharge processes can be oxidized to $NO_2$ in air and then react with OH radicals to form nitric acid [39], as shown in the below formulae

$$NO + OH \rightarrow HNO_2 \tag{3.8}$$

and

$$NO_2 + OH \rightarrow HNO_3. \tag{3.9}$$

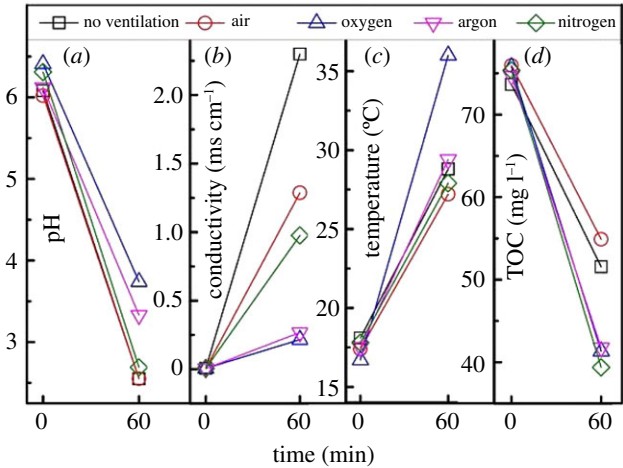

**Figure 6.** Different characteristic parameters of the solution after 60 min treatment.

$N_2^+$ and $H_2O$ can also form nitrate during $N_2$ discharge, as shown in formula (3.10) [40]

$$N_2^+ + 6H_2O \rightarrow 2NO_3^- + 12H^+ + 9e. \tag{3.10}$$

Therefore, the pH of the solution after degradation by air and $N_2$ discharge in figure 6a is approximately 2.5, which is significantly lower than that of the solution after Ar and $O_2$ discharging. After Ar discharging, the pH of the solution is 3.3, and the pH of the solution after $O_2$ discharging is 3.7. When Ar and $O_2$ are ventilated, no nitric acid is formed in the solution, and the pH is lowered maybe because of the formation of carboxylic acids.

As shown in figure 6b, the conductivity of the aniline solution significantly increased after 60 min treatment. This is because the aniline is degraded into small molecules or ions during the treatment. After 60 min treatment, the conductivity of the solution with oxygen ventilation is 0.22 mS cm$^{-1}$, argon is 0.27 mS cm$^{-1}$, no ventilation is 2.30 mS cm$^{-1}$, air is 1.29 mS cm$^{-1}$ and $N_2$ is 0.98 mS cm$^{-1}$. According to the analysis of the pH value, the formation of nitric acid and nitrous acid in the solution may have led to an increase in the solution conductivity in the case of no ventilation, air ventilation and $N_2$ ventilation.

The analysis of the solution temperature, as shown in figure 6c, indicates that the temperature of the solution significantly increased after 60 min treatment. This is because a part of the energy during the discharging is dissipated in the form of thermal energy and can heat the solution. The solution temperature increased to 36°C after 60 min treatment with $O_2$ discharge, and the temperature of the solution is approximately 28°C when the other gases are introduced. After being calculated by formula (2.1), the energy density when the reactor is ventilated with $O_2$ is 185 J l$^{-1}$, in other cases is about 100 J l$^{-1}$.

The TOC results shown in figure 6d indicate that the TOC of the solution decreases significantly after 60 min treatment under all the discharging atmospheres, which means that the aniline is mineralized after streamer corona plasma treatment. Figure 6d shows that the solution TOC is 51.6 and 54.89 mg l$^{-1}$ when no gas is ventilated and air is ventilated, and the solution TOC is approximately 40 mg l$^{-1}$ when the $O_2$, $N_2$ and Ar are ventilated. As shown in figure 3, although the aniline degradation in the solution is faster when no gas or the air is ventilated, it can be seen from the TOC analysis that the aniline mineralization is incomplete, which indicates that only the aniline ring structure may have been broken.

## 3.5. Mechanism of the aniline degradation

According to the experimental results, the degradation process of aniline can be obtained. It is estimated that the degradation products of aniline by gaseous streamer corona plasma can be divided into two series of nitrophenol and aminophenol. A schematic diagram of the degradation process of aniline is shown in figure 7.

In figure 7a, when air, $N_2$ or Ar is introduced, the degradation intermediate of aniline is nitrophenol, mainly in the para- and ortho- formations, while the productions of m-nitrophenol are extremely low. When Ar is discharged, the yield of nitrophenol is less than that of $N_2$ or air discharge because the

**Figure 7.** Diagram of aniline degradation processes.

gas phase products are relatively simple when Ar gas is discharged, and by-products are not easily generated. The *p*-nitrophenol in the solution can form *p*-benzoquinone under the attack of hydroxyl radicals, as shown in figure 4*c*. The aniline, nitrophenol and *p*-benzoquinone in the solution could be further cleaved into carboxylic acids under the attack of active particles and are finally mineralized into carbon dioxide and water [38].

These active substances that cleave aniline and its by-products can be divided into two parts. The first part is produced by gas phase discharge and dissolves in water to directly participate in the reaction. As measured in §3.3, there are N, $N_2$, $(C^3\Pi_u)$, $N_2^+$ and high-energy electrons $e^*$, in the $N_2$ discharge; Ar, OH and high-energy electrons $e^*$ are generated during the Ar discharge; the air discharge mainly contains active substances generated by nitrogen, and active substances O, $O_3$ and OH, generated by $O_2$. More specifically, air discharge produced nitrogen oxides such as NO. The second active substance formation pathway is that by indirect reaction. As shown in formula (3.6), the impact of high-energy electrons on $H_2O$ could produce OH radicals and H radicals. As shown in formula (3.2), the impact of excited state Ar on water could also produce OH radicals and H radicals. In addition, both $H_2O_2$ and OH radicals generated in liquid phase discharge could degrade aniline [20].

In figure 7*b*, when $O_2$ and Ar are introduced, the degradation intermediates of aniline are the para- and ortho-structured aminophenol, and the production of aminophenol is less when Ar is introduced. The active substance for the cleavage of aniline and its by-products can also be divided into two parts. The first is the active substances produced by the gas phase discharge that dissolved in water, and the second process is that of indirect reaction. The discharge in $O_2$ produced a large amount of $O_3$. Due to the strongly ortho-positioned amino group of aniline, it is susceptible to electrophilic attack by $O_3$ to form aminophenol. In addition, aniline could form aminophenol under the action of OH radicals and $O_2$ in water [41].

In summary, in the gaseous streamer corona plasma degradation process of the aniline solution, when Ar is introduced, the fewer by-products are generated, although the degradation rate of aniline in the Ar gas (figure 3) is slightly slower than that of the other gases, but the TOC of the solution was very low after 60 min treatment under Ar atmosphere, as shown in figure 6*d*, indicating that aniline could be effectively mineralized into inorganic products when Ar was introduced. When $O_2$ was introduced, the by-product of aniline degradation is aminophenol, but the degradation rate of aniline (figure 3) is fast, and the TOC of the solution is low after degradation. When $N_2$ is introduced, the by-product of aniline degradation is nitrophenol, which can be converted into *p*-benzoquinone during discharge. Although the TOC of the final solution is low, a large amount of nitric acid or nitrous acid is produced, resulting in a lower pH and a higher conductivity than other gas conditions. When the solution is degraded by air discharge, due to the complex composition of the gas phase, its by-products are quite different, including the formation of nitrophenol and a large amount of nitric acid and nitrous acid, resulting in a lower pH of the solution. Although the degradation of aniline

(figure 3) is fast, the TOC of the solution is high after 60 min treatment, indicating that the aniline is not completely mineralized.

# 4. Conclusion

The gaseous streamer corona plasma can effectively degrade aniline in water under different gas atmosphere, and the initial concentration is $100 \, \mathrm{mg \, l^{-1}}$. The degradation was fastest when the reactor was not ventilated, and the degradation rate is 98.5% under 7.5 min treatment. While the degradation was slowest when Ar was ventilated, after treatment for 60 min, the degradation rate is 98.6%. Some active particles were detected during the discharge, such as Ar, OH, $N_2$, $N_2^+$ and N. In particular, NO was detected during air discharge. The NO and $N_2^+$ would produce $NO_3^-$; then generated nitric acid would affect the pH value of the solution. The intermediate product in aniline degradation by $N_2$ discharge is nitrophenol, and nitrophenol would be converted to $p$-benzoquinone. It is mainly due to the breaking of the aniline benzene ring by the active substances. The intermediate products in air discharge are nearly the same as those produced in $N_2$ discharge. The $O_2$ discharge could produce an intermediate product of aminophenol. The intermediate products in Ar discharge are in small amounts and the final mineralization effect is the best.

Data accessibility. Data available from the Dryad Digital Repository: https://doi.org/10.5061/dryad.q573n5thd [42].

Authors' contributions. Y.L. carried out the degradation laboratory work, participated in data analysis, participated in the design of the study and drafted the manuscript; Zha.L. carried out the degradation laboratory work; Zhe.L. carried out the reactor design; S.H. carried out the emission spectrum experiment; S.Z. carried out the sample analysis; K.Y. conceived the study; A.Z. conceived of the study, designed the study, coordinated the study and helped draft the manuscript. All authors gave final approval for publication.

Competing interests. We declare we have no competing interests.

Funding. This work was funded by The National Natural Science Foundation of China (grant no. 51377145). Open Access funding provided by the Max Planck Society.

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
