## [Peer Review File · Royal Society Open Science]

Review History

RSOS-202314.R0 (Original submission)

Review form: Reviewer 1 (Nan Jiang)

Is the manuscript scientifically sound in its present form?

Yes

Are the interpretations and conclusions justified by the results?

Yes

Is the language acceptable?

Yes

Do you have any ethical concerns with this paper?

No

Have you any concerns about statistical analyses in this paper?

No

Recommendation?

Accept with minor revision (please list in comments)

Comments to the Author(s)

The manuscript titled "Degradation of Aniline in Water by a Pulsed Streamer Corona Plasma Using Different Gases along the Water Surface" describes the effects and influencing factors in degrading aniline with different gases, including air, oxygen, argon and nitrogen by a corona plasma. The formation of active species was detected. And the byproducts were also diagnosed. The obtained results were reliable, and it appears to be organized well. However, some of the statements in this manuscript are needed for further clarification.

Specific comments :

1. Introduction. Some other methods on aniline degradation or removal should be introduced. I recommend the authors to be more generous in the introduction, and also to use more references.
2. It is not clear the condition of no-venting. it means that there is ambient air? please clarify.
3. Energy density calculation should be moved to part 2 methods.
4. The Conclusions should be refined.

Summary: A minor revision is suggested before acceptance.

Review form: Reviewer 2

Is the manuscript scientifically sound in its present form?

Yes

Are the interpretations and conclusions justified by the results?

Yes

Is the language acceptable?

Yes

Do you have any ethical concerns with this paper?

No

Have you any concerns about statistical analyses in this paper?

No

Recommendation?

Accept with minor revision (please list in comments)

Comments to the Author(s)

This paper mainly investigated the effects and influencing factors in degrading of aniline in water with gaseous streamer corona plasma. In general, the experiments is comprehensive and the data is impressive and attractive. From this point of view, I recommend it with minor revision.

However, the expressions and structure should be further improved for this manuscript. Specific comments are as follows:

1. The abstract should clear point out the main results of your research and include some quantitative data.
2. The degradation kinetic model should be fitted in figures 3.
3. Past tense should be used in the manuscript.
4. ppm should be changed to mg/L.
5. Both the sentences and grammar should be further improved.

Decision letter (RSOS-202314.R0)

Dear Dr Li:

Title: Degradation of Aniline in Water With Gaseous Streamer Corona Plasma
Manuscript ID: RSOS-202314

Thank you for submitting the above manuscript to Royal Society Open Science. On behalf of the Editors and the Royal Society of Chemistry, I am pleased to inform you that your manuscript will be accepted for publication in Royal Society Open Science subject to minor revision in accordance with the referee suggestions. Please find the reviewers' comments at the end of this email.

The reviewers and handling editors have recommended publication, but also suggest some minor revisions to your manuscript. Therefore, I invite you to respond to the comments and revise your manuscript.

Because the schedule for publication is very tight, it is a condition of publication that you submit the revised version of your manuscript before 10-Mar-2021. Please note that the revision deadline will expire at 00.00am on this date. If you do not think you will be able to meet this date please let me know immediately.

Kind regards,
Dr Laura Smith
Publishing Editor, Journals

RSC Associate Editor:
Comments to the Author:
(There are no comments.)

RSC Subject Editor:
Comments to the Author:
(There are no comments.)

Reviewer comments to Author:
Reviewer: 1

Comments to the Author(s)

The manuscript titled "Degradation of Aniline in Water by a Pulsed Streamer Corona Plasma Using Different Gases along the Water Surface" describes the effects and influencing factors in degrading aniline with different gases, including air, oxygen, argon and nitrogen by a corona plasma. The formation of active species was detected. And the byproducts were also diagnosed. The obtained results were reliable, and it appears to be organized well. However, some of the statements in this manuscript are needed for further clarification.

Specific comments :

1. Introduction. Some other methods on aniline degradation or removal should be introduced. I recommend the authors to be more generous in the introduction, and also to use more references.
2. It is not clear the condition of no-venting. it means that there is ambient air? please clarify.
3. Energy density calculation should be moved to part 2 methods.

4. The Conclusions should be refined.

Summary: A minor revision is suggested before acceptance.

Reviewer: 2

Comments to the Author(s)

This paper mainly investigated the effects and influencing factors in degrading of aniline in water with gaseous streamer corona plasma. In general, the experiments is comprehensive and the data is impressive and attractive. From this point of view, I recommend it with minor revision.

However, the expressions and structure should be further improved for this manuscript. Specific comments are as follows:

1. The abstract should clear point out the main results of your research and include some quantitative data.
2. The degradation kinetic model should be fitted in figures 3.
3. Past tense should be used in the manuscript.
4. ppm should be changed to mg/L.
5. Both the sentences and grammar should be further improved.

Author's Response to Decision Letter for (RSOS-202314.R0)

See Appendix A.

Decision letter (RSOS-202314.R1)

Dear Dr Li:

Title: Degradation of Aniline in Water With Gaseous Streamer Corona Plasma
Manuscript ID: RSOS-202314.R1

It is a pleasure to accept your manuscript in its current form for publication in Royal Society Open Science. The chemistry content of Royal Society Open Science is published in collaboration with the Royal Society of Chemistry.

RSC Associate Editor
Comments to the Author:
(There are no comments.)

Reviewer(s)' Comments to Author:

Appendix A

Response to Referees

Response to Reviewer: 1

1. Some other methods on aniline degradation such as bioremediation, oxidation, adsorption and advanced oxidation processes have been introduced in the Introduction.
2. The condition of no-venting has been clarified in part 2.2, it means that there is ambient air but no gas flow.
3. Energy density calculation have been moved to part 2.
4. The Conclusion has been re-edited and refined.

Response to Reviewer: 2

1. The abstract has been re-edited and some quantitative data have been listed.
2. The degradation kinetic model have been fitted in figures 3(b).
3. Past tense have been used in the manuscript.
4. The ppm have been changed to mg/L in the Figure 6.
5. The sentences and grammar have been revised as correct as possible.